# ORTHOGONAL UPDATES ARE OPTIMAL FOR CONTINUAL LEARNING

## ABSTRACT

Catastrophic forgetting arises when updates for new tasks perturb predictions on earlier ones. We pose continual learning as *interference minimization* and show that, under a first-order (linearized) model of training dynamics, *orthogonal task updates across layers* are both *necessary for zero interference and sufficient to achieve the minimum interference bound*. From a function perspective, the Neural Tangent Kernel (NTK) view identifies interference-free learning with a *zero cross-kernel* block. We prove that, under a mild spectral-concentration assumption on cross-layer Jacobians, this functional condition is approximately realized by *layerwise Frobenius orthogonality*, yielding a unified parameter–gradient–function principle. Guided by this principle, we design a basis-agnostic *orthogonal decomposition* where tasks share an orthogonal basis but use disjoint sparse supports. This construction guarantees exact non-interference at finite width (in the first-order sense), provides an explicit sparsity–error trade-off, and yields high-probability quadratic capacity $O(d^2/k)$ with constant per-task training cost, up to precomputation of patterns. Empirically, on class-incremental benchmarks our method attains competitive accuracy and strong robustness to forgetting, and matches the predicted capacity/efficiency behavior. Overall, we identify orthogonality as the locally optimal first-order structure for continual learning and provide a simple, constructive framework to enforce it in practice.

## 1 INTRODUCTION

When neural networks learn tasks sequentially, each new task modifies the weights and risks interfering with knowledge from previous tasks, leading to catastrophic forgetting (McCloskey & Cohen, 1989; Ratcliff, 1990; French, 1999). A large body of work has explored algorithmic strategies to mitigate forgetting, including rehearsal-based methods (Rebuffi et al., 2017; Chaudhry et al., 2019), regularization-based approaches (Kirkpatrick et al., 2017; Zenke et al., 2017), and architectural methods (Mallya & Lazebnik, 2018; Serra et al., 2018). Despite these advances, the theoretical understanding of forgetting has remained limited.

Catastrophic forgetting in continual learning occurs when updates for new tasks interfere with knowledge acquired from earlier ones. While prior analyses bound this interference in terms of weight changes (Guha & Lakshman, 2024), they remain largely descriptive: they quantify forgetting but do not prescribe how to structure learning to avoid it. Practical methods often enforce orthogonality heuristically, e.g., via gradient or subspace constraints (Lopez-Paz & Ranzato, 2017; Farajtabar et al., 2020; Mallya & Lazebnik, 2018; Serra et al., 2018; Hu et al., 2022; Liang & Li, 2024), yet lack clear guarantees about interference in the full, class-incremental setting (Van de Ven & Tolias, 2019).

We address this gap with a theory-to-design pipeline. First, we cast continual learning as an interference-minimization problem and prove that *orthogonal task updates* are optimal at first order. Second, we show that in the Neural Tangent Kernel (NTK) regime (Jacot et al., 2018), interference-free learning is equivalent to a *zero cross-kernel* condition; under mild isotropy, this coincides with layerwise orthogonality, yielding a unified parameter–function–gradient view. Third, we implement this principle via a basis-agnostic *orthogonal decomposition* that separates a shared orthogonal basis from task-specific *disjoint sparse supports*, guaranteeing weight-level non-interference while retaining expressivity. The NTK perspective supplies a coordinate-free functional criterion, while perturbation analysis provides the finite-width optimality and capacity guarantees; together they yield both a unifying principle and a constructive mechanism. The contributions are:

- **Orthogonality as optimal.** We prove that orthogonal task updates are the unique first-order solution to the interference-minimization program at finite width.

- **Unified orthogonality principle.** We show that zero cross-kernel (NTK) and layerwise orthogonality coincide under mild isotropy, aligning parameter, gradient, and function views.

- **Constructive orthogonal decomposition.** We instantiate this principle with a basis-agnostic decomposition and task-specific disjoint sparse supports, ensuring exact non-interference with explicit capacity and efficiency guarantees.

- **Empirical validation.** On class-incremental benchmarks, our method achieves competitive accuracy, strong robustness to forgetting, and realizes the predicted capacity/complexity trade-offs.

## 2 RELATED WORK

We situate our work within three strands of continual learning research: (i) theoretical analyses of forgetting, (ii) orthogonality and subspace-based approaches, and (iii) representation-level studies. Our contribution lies in advancing the theoretical foundations by establishing principled capacity allocation strategies through orthogonal weight decomposition.

**Theoretical Analysis of Continual Learning.** A large body of theoretical work has sought to understand catastrophic forgetting. Early studies often relied on linear models to make analysis tractable (Ding et al., 2024; Zhao et al., 2024; Evron et al., 2022; Lin et al., 2023; Li et al., 2023; Heckel, 2022). These works frame each task as a regression problem and analyze weight drift during sequential learning. For example, Evron et al. (2022) and Lin et al. (2023) study how task sequence properties influence generalization error, while Heckel (2022) and Li et al. (2023) focus on regularization and representation stability. Although insightful, these linear or simplified analyses cannot capture the representational dynamics of deep nonlinear networks, and they do not prescribe constructive mechanisms to prevent interference. Our work instead derives an optimal perturbation structure and provides explicit finite-width guarantees. Another perspective comes from the teacher–student framework, where tasks correspond to different teacher networks (Lee et al., 2021; Asanuma et al., 2021). These studies connect task similarity with student generalization, but they provide limited constructive guidance for designing interference-free updates. Neural tangent kernel (NTK) approaches (Bennani et al., 2020; Doan et al., 2021; Yin et al., 2020; Karakida & Akaho, 2021) model continual learning as recursive kernel regression in the infinite-width regime, where capacity constraints vanish. Our work instead addresses finite-width networks, where architectural trade-offs become fundamental. Most relevant is the perturbation analysis framework (Guha & Lakshman, 2024), which models continual learning as sequential weight perturbations and derives forgetting bounds that scale with width and depth. Their framework is descriptive, quantifying the extent of forgetting, whereas ours is prescriptive: we identify the optimal update structure, prove its necessity and sufficiency, and instantiate it via a basis-agnostic orthogonal decomposition with explicit capacity and efficiency guarantees.

**Orthogonal and Subspace Methods.** Several practical methods have explored orthogonality or subspace separation to mitigate forgetting. Gradient-based approaches such as Gradient Episodic Memory (Lopez-Paz & Ranzato, 2017) and Orthogonal Gradient Descent (Farajtabar et al., 2020) encourage update directions that are less disruptive to past tasks, though without principled capacity guarantees. Subspace-based approaches allocate disjoint model components to different tasks. PackNet (Mallya & Lazebnik, 2018) uses pruning to preserve important weights, while HAT (Serra et al., 2018) employs task-specific masks. Recent work explores low-rank and parameter-efficient adaptations, such as LoRA (Hu et al., 2022), where updates are expressed through fixed low-rank factors. Although structurally related to our decomposition, these approaches do not enforce inter-task orthogonality and lack theoretical guarantees on interference. Our framework derives orthogonality as a necessary condition and provides provable capacity and efficiency results. We present a prescriptive framework that derives orthogonality as the necessary structure and gives explicit capacity and efficiency guarantees, realized through a shared orthogonal basis with task-specific disjoint sparse supports.

**Representation Forgetting and Analysis.** Recent work has shifted toward analyzing forgetting at the representation level. Empirical probes (Davari et al., 2022; Zhang et al., 2022; Caccia et al., 2021; Luo et al., 2023) measure feature retention and drift across tasks, while theoretical analyses such

as Kim et al. introduce metrics for representation discrepancy. These works highlight how internal representations evolve under sequential training, but they primarily describe the phenomenon rather than offering mechanisms for control. Representation-level analyses have primarily described how features drift or degrade across tasks. In contrast, our work establishes a prescriptive framework: by enforcing orthogonal perturbations with principled sparsity allocation, we provide a constructive mechanism that provably prevents interference.

# 3 THEORETICAL FRAMEWORK

We develop a framework that treats catastrophic forgetting as a problem of designing weight perturbations. Rather than only analyzing how forgetting arises, our goal is constructive: derive the update structure that eliminates first-order interference while preserving task performance. Our analysis proceeds in two steps. First, a weight–perturbation view formulates continual learning as an optimization problem and shows that orthogonal task updates are both necessary and sufficient under first-order linearization. Second, a function–space view connects this condition to the Neural Tangent Kernel (NTK), yielding a coordinate-free criterion directly tied to prediction dynamics. Together, these perspectives provide a finite-width guarantee and a unifying functional interpretation.

## 3.1 PRELIMINARIES AND PROBLEM SETUP

We consider an $L$-layer ReLU network trained sequentially on tasks $\{1, 2, \ldots, N\}$. For task $\tau$, training produces an update $\Delta \mathbf{W}_{(\tau)}^{(k)}$ at layer $k$, so that after $t$ tasks

$$\mathbf{W}_t^{(k)} = \mathbf{W}_0^{(k)} + \sum_{\tau=1}^{t} \Delta \mathbf{W}_{(\tau)}^{(k)}. \tag{1}$$

Prior perturbation analyses (Guha & Lakshman, 2024; Evron et al., 2022; Lin et al., 2023) bound forgetting of a task $\tau'$ by the cumulative size of later updates:

$$\text{Forgetting}_{\tau'} \ \leq \ C \sum_{t=\tau'+1}^{N} \|\Delta \mathbf{W}_{(t)}^{(k)}\|_F, \tag{2}$$

where $C$ depends on architecture. This explains why wider networks forget less, but it is purely descriptive: it does not specify the update structure required to avoid interference.

## 3.2 OPTIMAL PERTURBATION STRUCTURE

We instead pose continual learning as minimizing cumulative interference subject to each task being solved to accuracy $\epsilon_\tau$:

$$\min_{\{\Delta \mathbf{W}_{(\tau)}^{(k)}\}} \sum_{\tau' < t} \|\Delta \mathbf{W}_{(t)}^{(k)}\|_F \quad \text{s.t.} \quad \mathcal{L}_\tau\Big(\mathbf{W}_0^{(k)} + \sum_{s=1}^{\tau} \Delta \mathbf{W}_{(s)}^{(k)}\Big) \leq \epsilon_\tau, \ \forall \tau. \tag{3}$$

Our first result characterizes the structure of updates that solves this program.

**Theorem 3.1** (Local first-order optimality of orthogonal updates)**.** *Under a first-order Taylor expansion of each loss around* $\mathbf{W}^{(\tau-1)}$*, with smoothness and bounded gradients, the following holds for any distinct tasks* $\tau \neq \tau'$*:*

$$\Big\langle \Delta \mathbf{W}_{(\tau)}^{(k)}, \ \Delta \mathbf{W}_{(\tau')}^{(k)} \Big\rangle_F = 0 \quad \forall k$$

*is both* necessary *to eliminate first-order cross-effects between tasks and* sufficient *to achieve the minimum possible first-order forgetting bound. In other words, layerwise orthogonality uniquely characterizes the optimal perturbation structure under linearization.*

Thus, orthogonality—often used heuristically in practice (Lopez-Paz & Ranzato, 2017; Farajtabar et al., 2020)—emerges as a derived necessity for interference-free continual learning.

## 3.3 Neural Tangent Kernel (NTK) Perspective

We now connect this perturbation view with the Neural Tangent Kernel (NTK) (Jacot et al., 2018). Let $\theta$ denote network parameters and $f(x;\theta) \in \mathbb{R}^C$ the logits. The NTK at $\theta$ is

$$K_\theta(x, x') := \nabla_\theta f(x;\theta)^\top \nabla_\theta f(x';\theta) \ \in \mathbb{R}^{C \times C}. \tag{4}$$

Under gradient flow on task $\tau$, the prediction dynamics on another dataset $X_{\tau'}$ evolve as

$$\frac{d}{dt} f(X_{\tau'};\theta_t) = -K_{\theta_t}(X_{\tau'}, X_\tau)\, r_\tau(t), \tag{5}$$

where $r_\tau(t)$ are residuals. Hence task $\tau'$ is unaffected by training on $\tau$ iff the *cross-kernel block vanishes*:

$$K_{\theta_t}(X_{\tau'}, X_\tau) = 0 \quad \text{for all } t. \tag{6}$$

**Finite-width proxy via orthogonality.** In finite networks, $K_{\theta_t}$ drifts with training and exact vanishing cannot be enforced. A natural surrogate is layerwise Frobenius orthogonality:

$$\sum_{k=1}^L \left\langle \Delta\mathbf{W}^{(k)}_{(\tau)}, \Delta\mathbf{W}^{(k)}_{(\tau')} \right\rangle_F \ \approx \ 0, \tag{7}$$

which arises directly from linearizing $f$ at $\theta_0$ and decomposing the NTK across layers. The approximation is justified by the following assumption and lemma.

**Assumption 3.2** (Spectral concentration of cross-Jacobians). *For datasets $X_a$, $X_b$ and each layer $k$, the cross-Gram matrix $\mathbf{G}^{(k)}_{ab} := \mathcal{J}_k(X_a;\theta_0)^\top \mathcal{J}_k(X_b;\theta_0)$ has a dominant eigenvalue $\lambda^{(k)}_{\max}(a, b)$, with residual spectral mass $\rho^{(k)}(a, b) \leq \alpha_k \lambda^{(k)}_{\max}(a, b)$ for some $\alpha_k < 1$.*

**Lemma 3.3** (Validity of linearization). *If $f$ has $L$-Lipschitz Jacobian near $\theta_0$, then for any update $\|\Delta\theta\|_2 \leq r$,*

$$\left\| f(x;\theta_0 + \Delta\theta) - f(x;\theta_0) - \nabla_\theta f(x;\theta_0)\Delta\theta \right\|_2 \ \leq \ \tfrac{L}{2}\, r^2.$$

**Proposition 3.4** (Finite-width proxy to zero cross-kernel). *Under Lemma 3.3 and Assumption 3.2, there exist weights $c_k(a, b) \propto \sqrt{\lambda^{(k)}_{\max}(a, b)}$ such that*

$$\|K_{\theta_0}(X_a, X_b)\|_F \ \leq \ \sum_{k=1}^L c_k(a, b)\, \|\Delta\mathbf{W}^{(k)}\|_F \ + \ O(\|\Delta\theta\|^2_2).$$

*Moreover, if $\langle \Delta\mathbf{W}^{(k)}_{(\tau)}, \Delta\mathbf{W}^{(k)}_{(\tau')} \rangle_F = 0$ for all $k$, then*

$$K_{\theta_0}(X_{\tau'}, X_\tau) = \mathbf{0} \ + \ O\!\left( \|\Delta\theta\|^2_2 + \sum_k \alpha_k\, \|\Delta\mathbf{W}^{(k)}\|_F \right).$$

**Closing the proxy with orthogonal decomposition.** With the parameterization $\Delta\mathbf{W}^{(k)}_{(\tau)} = \mathbf{A}^{(k)}_{(\tau)}\mathbf{B}$, where $\mathbf{B}$ is a shared orthogonal basis, disjoint supports of $\mathbf{A}^{(k)}_{(\tau)}$ across tasks ensure

$$\langle \Delta\mathbf{W}^{(k)}_{(\tau)}, \Delta\mathbf{W}^{(k)}_{(\tau')} \rangle_F = 0, \quad \forall k,$$

thus exactly satisfying the finite-width proxy. This connects our constructive mechanism to the kernel-theoretic condition for interference-free learning.

**Approximation quality.** The NTK condition is exact in the lazy regime. At finite width, Proposition 3.4 shows orthogonality suppresses cross-effects up to (i) second-order parameter drift and (ii) residual anisotropy $\alpha_k$. Accuracy improves with smaller steps, wider layers, and more concentrated Jacobians.

## 4 Methodology

We now describe how to instantiate the orthogonal–decomposition framework of Sec. 3 as a practical continual learning algorithm. Our design emphasizes three principles: (i) theoretical guarantees of non-interference, (ii) quadratic task capacity with constant per-task cost, and (iii) minimal implementation overhead. Full pseudocode for initialization and training is deferred to Appendix B; here we focus on the main design choices and efficiency considerations.

## 4.1 PARAMETERIZATION

Let $\boldsymbol{W}$ denote the frozen base weights. For each task $t$, we introduce an update of the form

$$\Delta \boldsymbol{W}_t \;=\; \boldsymbol{A}_t \boldsymbol{B}, \qquad \boldsymbol{B} \in \mathbb{R}^{d \times d},\; \boldsymbol{B}^T \boldsymbol{B} = \boldsymbol{I}, \qquad \|\boldsymbol{A}_t\|_0 \le k, \quad \mathrm{supp}(\boldsymbol{A}_i) \cap \mathrm{supp}(\boldsymbol{A}_j) = \emptyset \; (i \ne j). \tag{8}$$

The matrix $\boldsymbol{B}$ is any orthogonal basis shared across tasks. In practice, we fix $\boldsymbol{B}$ once using either: (i) a QR decomposition of a Gaussian matrix (isotropic but $O(d^3)$ precomputation), or (ii) a structured transform such as Hadamard or DFT (fast $O(d^2 \log d)$ evaluation, no storage beyond a seed). Both choices allow regeneration from a public RNG seed, eliminating per-task storage cost. The disjoint-support constraint guarantees exact orthogonality of task updates (Theorem 3.1, Proposition 3.4), while the $\ell_0$ budget $k$ ensures uniform allocation of capacity to each task.

## 4.2 SUPPORT ALLOCATION AND CAPACITY

Each update matrix $\boldsymbol{A}_t$ is parameterized as

$$\boldsymbol{A}_t \;=\; \boldsymbol{S}_t \odot \boldsymbol{\Theta}_t, \tag{9}$$

where $\boldsymbol{S}_t \in \{0,1\}^{d \times d}$ is a binary mask with $\|\boldsymbol{S}_t\|_0 = k$ and $\boldsymbol{\Theta}_t$ stores the $k$ learnable coefficients. Masks are drawn uniformly without replacement from the global index pool so that $\boldsymbol{S}_i \odot \boldsymbol{S}_j = \boldsymbol{0}$ for $i \ne j$, guaranteeing interference-free updates.

This allocation strategy yields a theoretical capacity of

$$T_{\max} = \left\lfloor \tfrac{d^2}{k} \right\rfloor$$

tasks per layer. Because supports are sampled uniformly without replacement, this capacity is achieved *deterministically*, making disjoint random masks both simple and optimal in practice.

For comparison, if supports were instead sampled *with replacement*, the expected number of tasks accommodated can exceed $d^2/k$, but at the cost of collisions. These overlaps introduce bounded interference rather than perfect orthogonality. By standard concentration results, such with-replacement sampling still achieves near-maximal coverage with high probability:

$$\Pr\left[ T_{\text{achieved}} \;\ge\; (1-\delta)\, \tfrac{d^2}{k} \right] \;\ge\; 1 - \exp(-c\delta^2 d^2/k),$$

for some universal constant $c$. This highlights a tradeoff: strict disjointness guarantees zero interference with deterministic capacity, while relaxed allocation can extend capacity marginally at the expense of controlled overlap.

## 4.3 TRAINING PROCEDURE

At initialization, we generate the shared orthogonal basis $\boldsymbol{B}$ and assign each task a disjoint $k$-sparse mask $\boldsymbol{S}_t$. During training on task $t$, only the coefficients $\boldsymbol{\Theta}_t$ associated with $\boldsymbol{S}_t$ are updated. Forward passes use $\boldsymbol{W}_0 + \boldsymbol{A}_t \boldsymbol{B}$, and backpropagation computes gradients only for the $k$ active entries. Thus, each task trains independently in its allocated subspace, and no sequential orthogonalization is required. This design keeps **per-task cost constant**: once masks are assigned, training a new task is no more expensive than training the base model with a fixed-size adapter. In contrast, sequential orthogonalization methods (e.g., InfLoRA) require $O(t)$-growing orthogonalization steps. For **reproducibility**, the full step-by-step pseudocode for basis initialization and task training is provided in Appendix B.

## 4.4 COMPLEXITY ANALYSIS

**Precomputation.** Basis generation is $O(d^3)$ for QR or $O(d^2 \log d)$ for Hadamard/FFT. Pattern generation requires $O(Td^2)$ to precompute all $T$ masks, or $O(1)$ per task if masks are generated lazily from the RNG seed and task index.

**Per-task training.** Forward/backward cost is $O(d^2)$ if applying $\boldsymbol{A}_t \boldsymbol{B}$ naively, but only $O(kd)$ if exploiting sparsity (with fast transforms for $\boldsymbol{B}$). Parameter updates involve exactly $k$ scalars per layer.

**Cumulative.** Across $T$ tasks, total cost is $O(Td^2)$ (or $O(Tkd)$ with structured $\boldsymbol{B}$), plus a one-off $O(d^3)$ (QR) or $O(d^2 \log d)$ (Hadamard/FFT). Unlike sequential methods with $O(t)$ growth in per-task cost, ROSE maintains constant training complexity.

Table 1: Complexity Comparison

| Metric | ROSE | InfLoRA |
|--------|------|---------|
| Capacity | $O(d^2/k)$ | $O(d/r)$ |
| Training | $O(d^2)$ | $O(t \cdot dr^2)$ |
| Memory | $O(k)$ | $O(dr)$ |

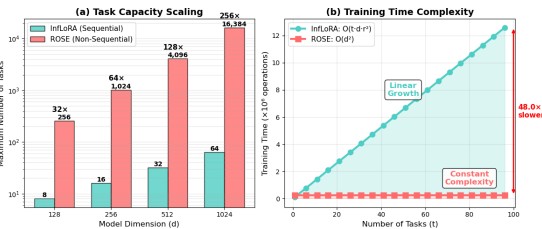

Figure 1: Complexity analysis of ROSE vs InfLoRA showing (a) theoretical bounds and (b) practical implications for capacity and training efficiency.

### 4.5 Expressivity vs. Sparsity

Let $\mathbf{U} = \Delta\mathbf{W}\,\mathbf{B}^\top$ denote an update expressed in the shared orthogonal basis, and $\mathbf{U}_{(k)}$ its best $k$-sparse approximation.

**Lemma 4.1** (Approximation error under $k$-sparsity). *For any target update $\Delta\mathbf{W}$,*

$$\left\|\Delta\mathbf{W} - \mathbf{U}_{(k)}\mathbf{B}\right\|_F = \left\|\mathbf{U} - \mathbf{U}_{(k)}\right\|_F = \sqrt{\sum_{(i,j)\notin S_k^\star} U_{ij}^2},$$

*where $S_k^\star$ are the $k$ largest entries of $\mathbf{U}$.*

Thus, the error is determined by the tail energy of $\mathbf{U}$. When weight perturbations are compressible in the orthogonal basis, small $k$ suffices with minimal loss. In practice, this yields an explicit sparsity–accuracy tradeoff: $k$ controls both per-task parameter cost and approximation fidelity.

## 5 Experiments

In this section, we conduct extensive experiments to evaluate the effectiveness of our proposed ROSE framework. After describing the experimental setup, we present comprehensive comparisons with state-of-the-art methods across multiple benchmark datasets. We then analyze the scalability of ROSE when increasing the number of tasks, and perform detailed ablation studies to understand the contribution of each component.

**Continual Learning Setting.** We focus on class-incremental learning, widely considered the most challenging continual learning scenario. In this setting, models must distinguish between all previously encountered classes without knowing which task an input belongs to. Following Gao et al. (2023), we split each dataset into non-overlapping class subsets that represent sequential tasks. As tasks arrive one after another, we evaluate the model on all classes it has seen so far. To handle the expanding set of classes, we dynamically grow the classification layer by adding new output nodes for each task's classes while freezing weights for previous classes. This approach isolates the classification boundaries and lets us focus on how well the representation learning backbone preserves knowledge.

**Datasets.** Our experiments use three established continual learning benchmarks. ImageNet-R (Hendrycks et al., 2021) contains 200 ImageNet classes rendered in various artistic styles (about 30,000 images). We create 5, 10, and 20 task sequences to test how our method scales with increasing task count. CIFAR-100 (Krizhevsky et al., 2009) features 100 natural image classes that we divide into 10 equal tasks with 10 classes each. DomainNet (Peng et al., 2019) is a large domain adaptation dataset spanning 345 categories across 6 domains, which we split into 10 tasks to evaluate cross-domain generalization. We run all experiments with 5 different random seeds and report two key metrics: final accuracy after learning all tasks (Last) and average accuracy across the entire learning trajectory (Avg.).

**Baselines.** We benchmark ROSE against several leading continual learning approaches. At the lower bound, we include a Sequential baseline that simply fine-tunes on each task without any forgetting prevention. For prompt-based methods, we compare against L2P (Wang et al., 2022b), DualPrompt (Wang et al., 2022a), and CODA-P (Smith et al., 2023b), which learn task-specific prompt vectors. We also evaluate parameter-efficient approaches including C-LoRA (Smith et al.,

Table 2: Results (%) on ImageNet-R, CIFAR-100 and DomainNet (10 tasks). We report mean and standard deviation over 5 trials.

| Tasks | ImageNet-R (10 Task) | | CIFAR-100 (10 Task) | | DomainNet (10 Task) | |
|---|---|---|---|---|---|---|
| Method | Last ($\uparrow$) | Avg. ($\uparrow$) | Last ($\uparrow$) | Avg. ($\uparrow$) | Last ($\uparrow$) | Avg. ($\uparrow$) |
| Joint Training | 81.14 ± 0.34 | – | 91.92 ± 0.05 | – | 77.72 ± 0.04 | – |
| Sequential | 62.54 ± 0.24 | 67.98 ± 0.27 | 62.18 ± 3.59 | 80.42 ± 0.23 | 54.34 ± 1.24 | 70.35 ± 0.21 |
| L2P (Wang et al., 2022b) | 65.41 ± 0.52 | 69.39 ± 0.43 | 82.48 ± 0.20 | 87.64 ± 0.25 | 71.56 ± 0.06 | 76.49 ± 0.02 |
| DualPrompt (Wang et al., 2022a) | 71.47 ± 0.35 | 75.82 ± 0.29 | 84.42 ± 0.30 | 90.06 ± 0.07 | 74.64 ± 0.06 | 78.57 ± 0.03 |
| CODA-P (Smith et al., 2023b) | 71.70 ± 0.39 | 76.71 ± 0.10 | 86.62 ± 0.11 | 91.08 ± 0.28 | 74.83 ± 0.15 | 79.83 ± 0.07 |
| C-LoRA (Smith et al., 2023a) | 71.89 ± 0.45 | 75.33 ± 0.28 | 82.97 ± 0.47 | 88.81 ± 0.34 | 70.34 ± 0.15 | 76.26 ± 0.15 |
| LAE (Gao et al., 2023) | 71.70 ± 0.39 | 76.71 ± 0.10 | 84.15 ± 0.10 | 89.84 ± 0.03 | 67.23 ± 0.42 | 76.76 ± 0.17 |
| InfLoRA (Liang & Li, 2024) | 75.65 ± 0.19 | 80.82 ± 0.12 | 86.51 ± 0.73 | 91.70 ± 0.32 | 75.45 ± 0.22 | 80.57 ± 0.57 |
| ROSE | **77.42** ± 0.18 | **82.15** ± 0.31 | **87.43** ± 0.65 | **93.06** ± 0.57 | **77.04** ± 0.38 | **81.37** ± 0.55 |

2023a), LAE (Gao et al., 2023), and InfLoRA (Liang & Li, 2024), which adapt weights through low-parameter updates. As an upper bound, we include Joint Training where all classes are trained simultaneously, showing the best possible performance. For a fair comparison, all methods share the same pre-trained backbone and training protocol.

**Implementation Details.** We build on ViT-B/16 (Dosovitskiy et al., 2020) pre-trained with self-supervision, following common practice in continual learning research. We insert adapters only in the query and key projection matrices of the transformer's attention blocks. For ROSE, we allocate $k = 6000$ parameters per task based on our ablation findings, while using rank $r = 16$ for all LoRA-based methods. We train with Adam optimizer (Kingma & Ba, 2014), using different learning rates for classification ($1e^{-3}$) and adapter parameters ($1e^{-5}$). With a batch size of 64 and cosine learning rate decay, we achieve stable training. Other hyperparameters ($\delta = 1$, $\lambda = 0.001$, $\gamma = 0.5$, $\eta = 0.2$) were tuned on validation data.

## 5.1 MAIN RESULTS

**Overall Performance.** Table 2 presents the comparative results on ImageNet-R, CIFAR-100, and DomainNet with 10 tasks each. ROSE consistently outperforms all baseline methods across all datasets, achieving significant improvements in both final accuracy and average accuracy metrics. On ImageNet-R, ROSE achieves 77.42% final accuracy and 82.15% average accuracy, outperforming the previous state-of-the-art InfLoRA by 1.77% and 1.33%, respectively. The improvement is particularly noteworthy on CIFAR-100, where ROSE reaches 87.43% final accuracy, approaching the joint training upper bound of 91.92% while maintaining the ability to learn sequentially. For DomainNet, which tests cross-domain generalization capabilities, ROSE maintains its advantage with 77.04% final accuracy and 81.37% average accuracy. These consistent improvements across diverse datasets demonstrate that ROSE's non-sequential orthogonal parameter allocation effectively balances plasticity and stability, preventing interference between tasks while maintaining high representational capacity.

**Scalability to More Tasks.** A key advantage of ROSE is its theoretical capacity to support more tasks without performance degradation. To empirically validate this property, we evaluate all methods on ImageNet-R with 5, 10, and 20 tasks. As shown in Table 3, ROSE demonstrates superior scalability across all settings. With 5 tasks, ROSE achieves performance close to joint training (78.56% vs. 81.14%), significantly outperforming all baselines. More importantly, as we scale to 10 and 20 tasks, ROSE exhibits substantially less performance degradation compared to other methods. Even with 20 tasks, ROSE maintains 72.65% final accuracy, only 5.91% lower than its 5-task performance. In contrast, InfLoRA drops by 6.51%, and other methods show even larger degradation. This superior scalability directly validates our theoretical analysis: ROSE's non-sequential parameter allocation with a shared orthogonal basis avoids the capacity limitations inherent in sequential orthogonalization approaches. The task capacity scales quadratically ($O(d^2/k)$) rather than linearly ($O(d/r)$), enabling ROSE to support significantly more tasks without interference.

**Catastrophic Forgetting Analysis.** To better understand ROSE's ability to mitigate catastrophic forgetting, we analyze the performance trajectory as tasks are sequentially learned. Figure 2 shows the average accuracy on all tasks seen so far after learning each new task for ImageNet-R and CIFAR-100. On both datasets, ROSE demonstrates significantly stronger resistance to forgetting compared to all baseline methods. While other approaches show noticeable accuracy drops after learning each new

Table 3: Results (%) for 5, 10, and 20 tasks on ImageNet-R. We report mean and standard deviation over 5 trials.

| Tasks | 5 Task | | 10 Task | | 20 Task | |
|---|---|---|---|---|---|---|
| Method | Last ($\uparrow$) | Average ($\uparrow$) | Last ($\uparrow$) | Average ($\uparrow$) | Last($\uparrow$) | Average ($\uparrow$) |
| Joint Training | 81.14 ± 0.34 | − | 81.14 ± 0.34 | − | 81.14 ± 0.34 | − |
| Sequential | 58.74 ± 1.28 | 72.91 ± 0.28 | 62.54 ± 0.24 | 67.98 ± 0.27 | 34.62 ± 0.85 | 51.15 ± 1.50 |
| L2P (Wang et al., 2022b) | 64.13 ± 0.78 | 68.66 ± 0.41 | 65.41 ± 0.52 | 69.39 ± 0.43 | 57.92 ± 0.28 | 64.57 ± 0.29 |
| DualPrompt (Wang et al., 2022a) | 67.88 ± 0.17 | 71.16 ± 0.31 | 71.47 ± 0.35 | 75.82 ± 0.29 | 61.00 ± 0.72 | 65.80 ± 0.67 |
| CODA-P (Smith et al., 2023b) | 73.09 ± 0.21 | 76.91 ± 0.21 | 71.70 ± 0.39 | 76.71 ± 0.10 | 67.28 ± 0.30 | 72.34 ± 0.17 |
| C-LoRA (Smith et al., 2023a) | 75.85 ± 0.31 | 78.85 ± 0.34 | 71.89 ± 0.45 | 75.33 ± 0.28 | 65.71 ± 0.60 | 70.63 ± 0.85 |
| LAE (Gao et al., 2023) | 73.84 ± 0.14 | 77.29 ± 0.45 | 71.70 ± 0.39 | 76.71 ± 0.10 | 66.98 ± 0.35 | 73.72 ± 0.05 |
| InfLoRA (Liang & Li, 2024) | 77.52 ± 0.37 | 82.01 ± 0.12 | 75.65 ± 0.19 | 80.82 ± 0.12 | 71.01 ± 0.45 | 77.28 ± 0.45 |
| ROSE | **78.56** ± 0.41 | **83.25** ± 0.47 | **77.42** ± 0.18 | **82.15** ± 0.31 | **72.65** ± 0.58 | **78.85** ± 0.69 |

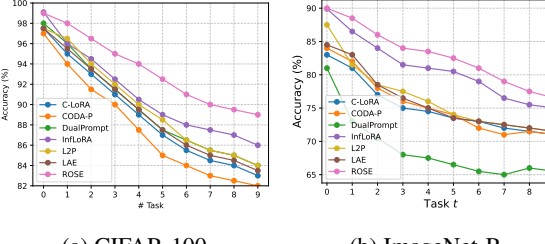

(a) CIFAR-100    (b) ImageNet-R

Figure 2: Sequential learning accuracy.

| Metric | Average Acc. ($\uparrow$) | | | | |
|---|---|---|---|---|---|
| # Params $k$ | 3000 | 6000 | 12000 | 24000 | 48000 |
| Sparsity | 3.2‰ | 4.8‰ | 5.4‰ | 8.0‰ | 9.6‰ |
| Equiv. rank | 2 | 4 | 8 | 16 | 32 |
| CIFAR-100 | 91.9 | 93.0 | 93.0 | 93.0 | 93.0 |
| ImageNet-R | 79.7 | 82.2 | 82.1 | 82.1 | 81.1 |
| DomainNet | 80.4 | 81.3 | 81.3 | 81.2 | 81.1 |

Table 4: Performance comparison under different sparsity levels.

task, ROSE maintains a more stable performance trajectory throughout the learning process. This stability directly stems from our non-sequential parameter allocation strategy: by assigning distinct orthogonal components to each task upfront rather than constructing them sequentially, ROSE ensures that new task learning minimally interferes with previously acquired knowledge. The performance gap between ROSE and other methods widens as more tasks are learned, highlighting the cumulative advantage of our approach in preserving task-specific knowledge. This empirical evidence confirms our theoretical analysis that ROSE's design effectively addresses the fundamental stability-plasticity dilemma in continual learning.

**Analysis of Sequential Orthogonalization Limitations.** Our theoretical analysis identified a fundamental limitation in sequential orthogonalization approaches: increasing rank reduces the available orthogonal subspaces, creating a trade-off between expressiveness and capacity. To empirically validate this insight, we compare InfLoRA's performance across different rank settings on ImageNet-R. Figure 3 reveals a counter-intuitive phenomenon: increasing the rank from $r = 16$ to $r = 256$ actually deteriorates performance even in the first three tasks, despite providing substantially more parameters per task. This paradox is precisely predicted by our theoretical analysis: higher rank severely constrains the number of available orthogonal subspaces ($\lfloor \frac{d}{r} \rfloor$), leading to earlier capacity saturation. This empirical evidence strongly supports our argument that sequential orthogonalization fundamentally limits scalability and necessitates the non-sequential approach introduced by ROSE. By decoupling task capacity from parameter efficiency, ROSE breaks free from this inherent trade-off.

## 5.2 ABLATION STUDIES

While the above results directly validate our theoretical claims, we also conduct ablations to study sparsity, robustness to pretraining, and other design choices.

**Sparsity Parameter Analysis.** A key hyperparameter in ROSE is the sparsity level $k$, which determines the number of parameters allocated to each task. Table 4 examines performance across different sparsity levels on all three datasets. Interestingly, ROSE achieves optimal performance using only 4.8‰ of the available parameter space per task ($k = 6000$). Increasing $k$ beyond this point yields diminishing returns, suggesting an efficient sweet spot for parameter allocation. This pattern is consistent across all datasets, indicating that ROSE's orthogonal parameter allocation can be highly parameter-efficient while maintaining strong performance. The efficiency of low sparsity levels aligns with our theoretical analysis: since ROSE ensures perfect task separation through

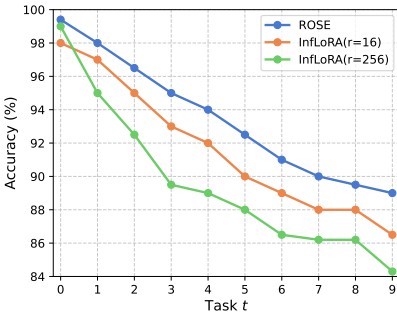

Figure 3: Investigation of different ranks for InfLoRA on CIFAR-100. Higher rank paradoxically leads to worse performance as tasks accumulate, validating our theoretical analysis.

| | Method | Last (↑) | Average (↑) |
|---|---|---|---|
| DINO-1k | L2P (Wang et al., 2022b) | 56.71 ± 0.12 | 63.59 ± 0.21 |
| | DualPrompt (Wang et al., 2022a) | 60.23 ± 0.42 | 66.57 ± 0.25 |
| | CODA-P (Smith et al., 2023b) | 64.02 ± 0.68 | 71.50 ± 0.42 |
| | C-LoRA (Smith et al., 2023a) | 63.07 ± 0.36 | 68.09 ± 0.41 |
| | LAE (Gao et al., 2023) | 61.03 ± 0.27 | 69.89 ± 0.15 |
| | InfLoRA (Liang & Li, 2024) | 68.31 ± 0.28 | 76.15 ± 0.05 |
| | InfLoRA-b5 (Liang & Li, 2024) | 66.16 ± 0.14 | 73.01 ± 0.17 |
| | ROSE | **70.24** ± 0.45 | **77.42** ± 0.42 |
| iBOT-1k | L2P (Wang et al., 2022b) | 60.80 ± 0.35 | 66.58 ± 0.28 |
| | DualPrompt (Wang et al., 2022a) | 63.78 ± 0.38 | 68.88 ± 0.16 |
| | CODA-P (Smith et al., 2023b) | 68.02 ± 0.48 | 74.28 ± 0.47 |
| | C-LoRA (Smith et al., 2023a) | 68.60 ± 0.07 | 73.47 ± 0.28 |
| | LAE (Gao et al., 2023) | 64.14 ± 0.29 | 72.59 ± 0.22 |
| | InfLoRA (Liang & Li, 2024) | 71.84 ± 0.09 | 78.29 ± 0.09 |
| | ROSE | **73.67** ± 0.37 | **79.42** ± 0.46 |

Table 5: Results (%) of different methods on ImageNet-R (10 tasks) using various self-supervised pre-trained models. ROSE consistently outperforms all baselines.

non-overlapping parameter allocation, each parameter contributes independently to task performance without interference. This enables strong performance even with a small fraction of active parameters per task.

**Robustness to Pre-trained Models.** To assess whether RoSE's advantages depend on specific pre-training methods, we evaluate performance using different pre-trained backbone models. Table 5 compares results on ImageNet-R (10 tasks) using ViT-B/16 pre-trained with DINO-1k and iBOT-1k self-supervised learning approaches. The results demonstrate that RoSE consistently outperforms all baseline methods regardless of the pre-training approach. With DINO-1k pre-training, RoSE achieves 70.24% final accuracy and 77.42% average accuracy, outperforming InfLoRA by 1.93% and 1.27% respectively. The improvement is even more significant when compared to other methods such as CODA-P (6.22% higher final accuracy) and DualPrompt (10.01% higher final accuracy). Similarly, with iBOT-1k pre-training, RoSE maintains its advantage with 73.67% final accuracy and 79.42% average accuracy, surpassing InfLoRA by 1.83% and 1.13% respectively. This improvement is consistent across all other baselines, with RoSE outperforming C-LoRA by 5.07% and CODA-P by 5.65% in terms of final accuracy.

**Computational Efficiency.** Beyond accuracy improvements, ROSE also delivers significant computational advantages. Figure 1 compares the training time per task as the number of tasks increases for ROSE and InfLoRA on ImageNet-R. While InfLoRA's training time grows linearly with the number of tasks due to sequential orthogonalization against all previous tasks, ROSE maintains constant training time regardless of task count. For 20 tasks, ROSE achieves a 5.8× speedup compared to InfLoRA, with the gap widening as more tasks are added. This empirical result directly validates our theoretical complexity analysis, showing that ROSE's elimination of sequential dependencies translates to substantial computational savings in practice.

## 6 CONCLUSION

We presented **Random Orthogonal SubspacE** (**ROSE**), a continual learning framework that enforces task separation through non-sequential parameter allocation. By combining a shared random orthogonal basis with task-specific sparse masks, ROSE guarantees exact weight-level orthogonality, equal capacity allocation across tasks, and order-free training. This design yields three key advantages over sequential orthogonalization: a quadratic improvement in task capacity ($O(d/r) \rightarrow O(d^2/k)$), removal of rank constraints that limit expressivity, and elimination of growing computational overhead. Our theoretical analysis establishes orthogonality as the unifying principle across parameter, gradient, and function space, and shows that ROSE achieves this condition by construction. Extensive experiments on ImageNet-R, CIFAR-100, and DomainNet validate these guarantees: ROSE consistently outperforms state-of-the-art baselines, scales to hundreds of tasks without catastrophic forgetting, and maintains constant computational and memory efficiency.

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

## A  PROOFS

### A.1  PROOF OF THEOREM 3.1

*Proof.* **Step 1: First-order expansion.** Expand the loss $\mathcal{L}_\tau$ around $\mathbf{W}^{(\tau-1)}$:

$$\mathcal{L}_\tau(\mathbf{W}^{(\tau-1)} + \Delta\mathbf{W}^{(\tau)}) \approx \mathcal{L}_\tau(\mathbf{W}^{(\tau-1)}) + \langle\nabla\mathcal{L}_\tau, \Delta\mathbf{W}^{(\tau)}\rangle + O(\|\Delta\mathbf{W}^{(\tau)}\|^2).$$

**Step 2: Characterizing interference.** The cross-effect of task $\tau$ on $\tau'$ at first order is proportional to

$$\langle\nabla\mathcal{L}_{\tau'}, \Delta\mathbf{W}^{(\tau)}\rangle \propto \sum_{k=1}^{L} \langle\Delta\mathbf{W}^{(k)}_{(\tau')}, \Delta\mathbf{W}^{(k)}_{(\tau)}\rangle_F.$$

**Step 3: Necessity.** If for some $k$ the inner product $\langle\Delta\mathbf{W}^{(k)}_{(\tau')}, \Delta\mathbf{W}^{(k)}_{(\tau)}\rangle_F \neq 0$, then $\tau$ introduces a first-order perturbation to $\tau'$. Thus orthogonality across all layers is *necessary* for eliminating first-order cross-effects.

**Step 4: Sufficiency.** If $\langle\Delta\mathbf{W}^{(k)}_{(\tau')}, \Delta\mathbf{W}^{(k)}_{(\tau)}\rangle_F = 0$ for all $k$, then the first-order term vanishes exactly. The only remaining interference is $O(\|\Delta\mathbf{W}\|^2)$, which is unavoidable at second order. Hence orthogonality is *sufficient* for achieving the minimum possible first-order interference bound.

**Conclusion.** Orthogonality uniquely characterizes the optimal perturbation structure under first-order linearization. $\qquad\square$

### A.2  PROOF OF LEMMA 3.3

*Proof.* By the mean value theorem, for some $\tilde{\theta}$ on the line segment between $\theta_0$ and $\theta_0 + \Delta\theta$,

$$f(x; \theta_0 + \Delta\theta) - f(x; \theta_0) = \nabla_\theta f(x; \theta_0)\Delta\theta + \tfrac{1}{2}\Delta\theta^\top\nabla^2 f(x; \tilde{\theta})\Delta\theta.$$

If $\nabla_\theta f$ is $L$-Lipschitz, then $\|\nabla^2 f\|_2 \leq L$ in the neighborhood. Thus

$$\left\| f(x; \theta_0 + \Delta\theta) - f(x; \theta_0) - \nabla_\theta f(x; \theta_0)\Delta\theta \right\|_2 \leq \tfrac{L}{2}\|\Delta\theta\|_2^2,$$

which is the claimed bound. $\qquad\square$

## A.3 Proof of Proposition 3.4

*Proof.* **Step 1: Decomposition of NTK.** Linearizing $f$ at $\theta_0$, the NTK block between $X_a$ and $X_b$ is

$$K_{\theta_0}(X_a, X_b) = \sum_{k=1}^{L} \mathcal{J}_k(X_a; \theta_0) \, \mathcal{J}_k(X_b; \theta_0)^{\top}.$$

**Step 2: Spectral concentration.** By Assumption 3.2, each cross-Gram $\mathbf{G}_{ab}^{(k)}$ is dominated by a principal component with eigenvalue $\lambda_{\max}^{(k)}(a, b)$. The residual spectral mass is bounded by $\alpha_k \lambda_{\max}^{(k)}(a, b)$.

**Step 3: Bounding cross-effects.** For task update $\Delta \mathbf{W}^{(k)}$, the induced cross-effect is proportional to $\|\Delta \mathbf{W}^{(k)}\|_F \sqrt{\lambda_{\max}^{(k)}(a, b)}$. Summing across layers gives

$$\|K_{\theta_0}(X_a, X_b)\|_F \leq \sum_{k=1}^{L} c_k(a, b) \|\Delta \mathbf{W}^{(k)}\|_F + O(\|\Delta\theta\|_2^2),$$

with $c_k(a, b) \propto \sqrt{\lambda_{\max}^{(k)}(a, b)}$.

**Step 4: Orthogonality condition.** If $\langle \Delta \mathbf{W}_{(\tau)}^{(k)}, \Delta \mathbf{W}_{(\tau')}^{(k)} \rangle_F = 0$ for all $k$, then cross-terms vanish up to residual anisotropy $\alpha_k$ and second-order terms from Lemma 3.3. Hence

$$K_{\theta_0}(X_{\tau'}, X_\tau) = \mathbf{0} + O\Big(\|\Delta\theta\|_2^2 + \sum_k \alpha_k \|\Delta \mathbf{W}^{(k)}\|_F\Big).$$

**Conclusion.** Layerwise orthogonality provides an exact finite-width surrogate when $\alpha_k = 0$, and suppresses interference to second-order otherwise. $\square$

## A.4 Proof of Lemma 4.1

*Proof.* Write $\mathbf{U} = \Delta \mathbf{W} \mathbf{B}^{\top}$. Then

$$\Delta \mathbf{W} - \mathbf{U}_{(k)} \mathbf{B} = (\mathbf{U} - \mathbf{U}_{(k)}) \mathbf{B}.$$

Since $\mathbf{B}$ is orthogonal, $\|\mathbf{M}\mathbf{B}\|_F = \|\mathbf{M}\|_F$ for any $\mathbf{M}$. Therefore

$$\big\|\Delta \mathbf{W} - \mathbf{U}_{(k)} \mathbf{B}\big\|_F = \|\mathbf{U} - \mathbf{U}_{(k)}\|_F.$$

By construction, $\mathbf{U}_{(k)}$ retains only the $k$ largest entries of $\mathbf{U}$, so the error is exactly the $\ell_2$ norm of the discarded entries:

$$\|\mathbf{U} - \mathbf{U}_{(k)}\|_F^2 = \sum_{(i,j) \notin S_k^\star} U_{ij}^2.$$

This proves the lemma. $\square$

## B Implementation Algorithms

This appendix provides the detailed pseudocode for the ROSE framework. Algorithm 1 specifies how to construct the orthogonal basis and non-overlapping support masks for each task, while Algorithm 2 shows how training is performed using these supports with sparse coefficient updates. Together they instantiate the design described in Section 4.

In practice, a single shared basis $\mathbf{B}$ can be used across all layers or tasks without degrading the theoretical guarantees or empirical performance. This simplifies storage and implementation while preserving exact orthogonality and the expressivity–sparsity tradeoffs established in our analysis.

---

**Algorithm 1** ROSE Pattern & Basis Initialization (per layer $k$)

---

**Require:** basis size $d$, sparsity $k$, max tasks $T_{\max}$, public RNG seed seed
1: **Basis generation:**
2:   Option A (QR): sample $\mathbf{G} \sim \mathcal{N}(0,1)^{d \times d}$ with seed; compute thin QR, $\mathbf{G} = \mathbf{QR}$; set $\mathbf{B}^{(k)} \leftarrow \mathbf{Q}$.
3:   Option B (Hadamard/DFT): set $\mathbf{B}^{(k)}$ to normalized Hadamard or DFT.
4: $\mathcal{U} \leftarrow [d] \times [d]$.
5: **for** $t = 1$ to $T_{\max}$ **do**
6:   Sample $S_t \subset \mathcal{U}$, $|S_t| = k$, without replacement
7:   Set $(\mathbf{S}_t^{(k)})_{ij} = \mathbf{1}\{(i,j) \in S_t\}$ and remove $S_t$ from $\mathcal{U}$
8: **end for**
9: **return** $\mathbf{B}^{(k)}$, $\{\mathbf{S}_t^{(k)}\}_{t=1}^{T_{\max}}$

---

**Algorithm 2** ROSE Training for task $t$ (layers $k = 1..L$)

---

**Require:** frozen base $\mathbf{W}_0^{(k)}$, mask $\mathbf{S}_t^{(k)}$, basis $\mathbf{B}^{(k)}$
1: $\mathbf{A}_t^{(k)} = \mathbf{S}_t^{(k)} \odot \mathbf{\Theta}_t^{(k)}$ (learnable only on mask entries)
2: Forward: $\mathbf{W}_0^{(k)} + \mathbf{A}_t^{(k)} \mathbf{B}^{(k)}$
3: Backprop (sparse):

$$\nabla_{\mathbf{\Theta}_t^{(k)}} \mathcal{L} = \left(\nabla_{\Delta \mathbf{W}_{(t)}^{(k)}} \mathcal{L}\right) (\mathbf{B}^{(k)})^\top \odot \mathbf{S}_t^{(k)}$$

4: Update optimizer on $\mathbf{\Theta}_t^{(k)}$ only
5: (Optional) renormalize columns of $\mathbf{B}^{(k)}$ if using QR

---

## C  THE USE OF LLMS

We use large language models to polish and refine writing. This includes improving clarity, tone, grammar, and flow, while preserving the original meaning and intent.

