# OpenReview forum: "Orthogonal Updates Are Optimal for Continual Learning"
_ICLR.cc/2026/Conference — Submitted to ICLR 2026_

### Official Review · Reviewer_mSgP · 2025-10-26

**Soundness:** 3
**Presentation:** 1
**Contribution:** 3
**Rating:** 4
**Confidence:** 3

**Summary:**

This paper investigates the problem of catastrophic forgetting from a theoretical perspective. The authors pose continual learning (CL) as an interference minimization problem. The paper's central claim is that, under a first-order (linearized) model of training dynamics, "orthogonal updates are optimal" for CL. The authors provide a theoretical argument, claiming that layerwise orthogonal task updates are both necessary and sufficient to minimize interference. They connect this parameter-space view to a function-space view via the Neural Tangent Kernel (NTK), arguing that zero cross-kernel interference is approximated by this layerwise orthogonality. Based on this principle, they design a method, ROSE, which uses a shared orthogonal basis and disjoint sparse supports for each task to enforce this orthogonality by construction.

**Strengths:**

1. The paper attempts to formalize the widely-used heuristic of orthogonal updates, aiming to provide a unifying theoretical principle that connects parameter-space, gradient-space, and function-space (NTK) views of interference. It proves that layerwise orthogonality is the unique first-order solution to an interference-minimization program and ties this to an NTK (function) condition; that gives both a principled insight and a coordinate-free interpretation.

2. This paper proposes a basis-agnostic decomposition (shared orthogonal basis + disjoint sparse supports) that yields deterministic capacity $O(d^2/k)$ and constant per-task cost, i.e., not just asymptotic handwaving but explicit capacity/complexity claims and implementation details.

**Weaknesses:**

1. I found the paper's core theoretical argument (Sec 3) exceptionally difficult to parse. The text is laden with jargon and dense mathematical formalism that, in my opinion, obscures rather than clarifies the main ideas. For a paper that aims to provide a new theoretical foundation, this lack of clarity is a significant obstacle. For instance, the connection between the NTK perspective and the final finite-width proxy in Sec 3.3 was not clearly articulated.

2. The paper invests significant effort in proving that orthogonal updates are "optimal" (in a limited, first-order sense). However, this theoretical formalization (Thm 3.1) feels like it is confirming a widely-held intuition (i.e., that non-interfering updates do not interfere) rather than providing a surprising new insight that pushes the field forward.

3. My main conceptual concern with the paper is its definition of "optimality." The authors equate "optimal for CL" with "minimal interference." This seems to fundamentally miss the point of the stability-plasticity dilemma. An update can have zero interference (perfect stability) but be a terrible update for learning the new task (zero plasticity). The paper's entire framework is built on solving for stability, but this is only half of the problem. A truly "optimal" CL method must co-optimize for both stability and plasticity, and this paper does not seem to address how to find an update that is also good for learning the new task.

4. The writing of this paper is poor. For example, sections often introduce notation before motivation, forcing readers to guess why certain variables or decompositions matter. Sentences are overly long and dense, filled with nested clauses. Many require multiple readings to parse. Additionally, there are frequent grammatical issues and awkward phrasing. The description of their algorithm sometimes skips practical details — for instance, how exactly the “shared orthogonal basis” is computed in finite width, or what hyperparameters are fixed during masking.

**Questions:**

1. For a paper whose main claims are theoretical, the exposition needs to be much clearer. Could the authors provide a simplified, intuitive explanation of the argument in Section 3.3?

2. I'm struggling to see the core theoretical novelty here, beyond a formal proof for an already-accepted heuristic. What, in the authors' view, is the most surprising or non-trivial consequence of their theoretical formalization?

3. The paper's premise seems to be that minimizing interference is the goal of CL. But isn't the goal to learn sequentially? How does the theory account for plasticity? Why should we accept a framework as "optimal" when it only solves for one half of the stability-plasticity trade-off?

---

### Official Review · Reviewer_c9xv · 2025-10-29

**Soundness:** 2
**Presentation:** 3
**Contribution:** 2
**Rating:** 4
**Confidence:** 3

**Summary:**

The authors are interested in the problem of catastrophic forgetting in the continual learning setting.  The first portion of the paper observes that (a quantity related to) forgetting is in some sense minimized when the updates corresponding to tasks are pairwise orthogonal.  This is phrased for the weight matrices of a neural network, and is also extended to the NTK regime.

Motivated by this theoretical observation, the authors propose a technique called RoSE, which is designed to avoid catastrophic forgetting by enforcing orthogonal updates.  The idea is rotate the weight matrices into a random basis and then to choose disjoint sets of coordinates in this basis, with each set corresponding to a task.  For each task, only the coordinates in the corresponding set are updated, so this enforces orthogonal updates between tasks. The authors then carry out experiments on a variety of datasets using 5,10 or 20 tasks.  The experimental results seem to suggest that RoSE outperforms the baselines in their experimental setup.

**Strengths:**

The experimental data looks relatively good and seems to show that RoSE outperforms similar benchmarks like Inflora with respect to measures of catastrophic forgetting.  The overall appraoch behind RoSE is also natural and well-motivated.  A slightly more obvious solution would be to partion the inputs/outputs of the layers into disjoint sets, but RoSE seems to improve on this, effectively by instead decomposing the weight matrices into matrices that are pairwise orthogonal with respect to the Frobenius inner product.

While there are issues with the theory (see below), the authors also make a reasonable attempt at theoretically justifying their algorithm.

**Weaknesses:**

Theorem 3.1 isn’t stated precisely.  I’m generally okay with a slightly informal statement if the intention is clear.  However here the proof is also informal, and so it’s hard to reconstruct what the underlying formal statement is meant to be. As written, Theorem 3.1 seems incorrect without at least some additional assumptions. It’s particularly confusing, because Theorem 3.1 is phrased locally.  Under first-order perturbations the loss function doesn’t change to zero-order, but the problem is phrased as a constrained optimization problem with respect to the values of the loss function.  In the proof, the constraint never seems to show up, so I’m not sure what the intention is, unless we’re meant to assume that the constraints are already satisfied locally (in which case the result is trivial since each update would be 0).  I imagine that this could be fixed with an appropriate statement, but the rigor here is sufficiently lacking that it makes me lose some trust in the rest of the paper.

RoSE is a nice idea, although I would imagine that its usefulness is somewhat limited in practice. Implementing something like RoSE would require planning ahead by effectively operating with many fewer parameters early in training. My guess would be that this is rarely a useful tradeoff, although I'm not an expert in the practical implementation of such systems so I could very well be wrong.

**Questions:**

Could the authors clarify the precise statment of Theorem 3.1?

RoSE operates by first rotating into a random basis (e.g. with a Hadamard for efficiency).  It would be nice to justify this step. How does this compare to just choosing disjoint supports in the standard basis?

The experiments seem to show that RoSE typically outperforms the benchmarks after zero tasks.  Why is this?

If the experiments don't take too long to run, it would be nice to see more than five trials per datapoint.  RoSE sometimes only wins by a fairly small margin, but yet it seems to do better in every experiment without exception.  This would be much more compelling with a large number of trials.  Relatedly, how were the parameters for the other methods chosen in order to have a fair comparison?

--- Presentation comments ---

Eq(1) – Might clarify the model.  Presumably we’re in the small perturbation regime so that the gradients are all calculated at the same point?

Eq (2) – what definition of forgetting is being used here?
“This explains why wider networks forget less” – why does it explain this?

Eq (3) – would be good to define $\mathcal{L}_{\tau}$ here or at least make a mention of it
This also seems to be a slightly imprecise formulation in the small perturbation regime where the loss function shouldn’t change.

Page 5 – Is W a $d\times d$ matrix? It wasn’t clear to me that it’s meant to be square until later down the page.

“scales to hundreds of tasks without catastrophic forgetting” – were there experiments with more than 20 tasks?

---

### Official Review · Reviewer_kPyx · 2025-11-01

**Soundness:** 3
**Presentation:** 1
**Contribution:** 2
**Rating:** 2
**Confidence:** 5

**Summary:**

This paper presents a CL algorithm, primarily for finetuning ViT models, that improves model CL capacity and efficiency.
The paper gives some theoretical analysis.
The realization of the proposed algorithm on ViT models are tested on numerous practical settings.

**Strengths:**

The practical result is impressive and the method is intuitive. With the efficiency improvement, it has the potential for a considerable practical impact on the CL + LLM community. If this paper were framed as a practical CL paper and written in such a way, I would be highly likely to recommend acceptance.

**Weaknesses:**

The paper, including the title, abstract, and Sections 1,2,3, is written as if the authors have proved a huge theoretical result.
Rather, in their theoretical part Section 3, Thm 3.1 is well-known and trivial, and Section 3.3 is vacuous and does not bring more messages than existing CL NTK analysis than [Bennani et al]. and [Doan et al.].

Furthermore, in the practical part, the paper avoids emphasizing that the proposed CL algorithm is better applied to Transformer-based models like LLMs and ViTs. My impression of the *real* contribution of this paper is that, compared to C-LoRA and InfLoRA, the random allocation of support before finetuning makes capacity, training, and memory efficiency better than the existing method. These benefits are highly likely to be dependent on the capacity of Transformer-based models without harming the performance. Note that the method is only verified on ViT. The improvements are highly unlikely to have been coming from the orthogonal updates, since the other LoRA-based methods also uses orthogonality.

**Questions:**

I suggest the authors to reformulate the paper. This could become a good empirical paper by adding solid analysis and ablation study on capacity and efficiency, not orthogonality.

Also, please avoid bashing theoretical CL works in the related works while painting a vacuous claim in title, abstract, intro, and Section 3. This is highly insulting to the theoretical CL community.

---

### Official Review · Reviewer_zn8f · 2025-11-03

**Soundness:** 1
**Presentation:** 1
**Contribution:** 1
**Rating:** 2
**Confidence:** 4

**Summary:**

This paper investigates catastrophic forgetting by modeling continual learning as an interference minimization problem. The authors derive a condition for zero forgetting based on a first-order approximation of the loss, concluding that layer-wise orthogonal updates are locally optimal. This theoretical principle is then connected to a fixed-NTK (linearized) perspective and implemented through a sparsity-based method that uses a shared orthogonal basis with disjoint task-specific supports, which is evaluated in the context of low-rank adapters.

**Strengths:**

The paper addresses the important challenge of catastrophic forgetting by attempting to derive a principled update structure from a theoretical standpoint. The goal of unifying the parameter, gradient, and function-space perspectives on interference is a valid and potentially interesting research direction.

**Weaknesses:**

* **Limited Theoretical Novelty:** The central theoretical claim—that orthogonal updates are optimal—is a direct and straightforward consequence of a first-order Taylor approximation of the loss. This derivation does not appear to represent a significant theoretical advancement, as it relies on a well-known linearization. The paper lacks a deep and critical analysis of the consequences of this simplification.
* **Fragile Assumptions:** The paper's entire theoretical framework rests on this first-order approximation (and the related fixed-NTK view). The authors rightly acknowledge (e.g., in Lemma 3.3) that these linear approximations break down easily in practice, as their validity depends on parameter norms. However, this admission is not followed by any empirical validation or discussion of the regime in which this approximation actually holds, calling into question the foundation of the proposed method.
* **Disconnect Between Theory and Method:** The final proposed method, which reduces to a simple sparsity-based update on a fixed basis, feels disconnected from the general theory. Such sparse update mechanisms are not novel and have been explored extensively in prior work (which is not adequately cited or compared against), often with more principled justifications. The paper fails to demonstrate what new, practical insight its theoretical derivation provides over these existing approaches.
* **Poor Presentation and Structure:** The paper is very difficult to follow. The theoretical framework is presented in a highly general, almost independent manner, while the specific context of the work—low-rank adapters—is confusingly withheld until the methodology section. This disconnect obscures the paper's true, and seemingly narrow, contribution. A general recommendation would be to first provide a high-level overview of the method and establish clear notation before presenting the detailed derivations.
* **Minor Formatting:** There are several minor but distracting formatting issues, such as inconsistent spacing in section titles.

**Questions:**

None

---

### Meta-Review · Area_Chair_Yc92 · 2025-12-05

**Summary:**

Probably the main concern raised by the reviewers is that the theoretical novelty contained in this paper is rather limited and "oversold" in the paper. Also concerns are raised regarding the clarity of the paper, and the precision of the theorems. All four reviewers recommend or support rejection, and no author rebuttal was provided. As AC, after going through the paper and the reviews, I do not see clear reasons to go against the unanimous advice of the reviewers, and I therefore recommend rejection.

**Reviewer Concerns:**

No rebuttal was provided.

**Reviewer Scores:**

Given that no rebuttal was provided, I think none of the reviewers would have changed their score.

---

### Decision · Program_Chairs · 2026-01-26

Reject